# Consistency of Central and Regional Planning in the Agricultural Sectors and the Factors Affecting It in Indonesia

**Wahyudi** [1] , **Imam Mujahidin Fahmid** [2] , **Darmawan Salman** [2],* **and Sultan Suhab** [3]

1   Doctoral Program of Development Studies, Graduate School, Hasanuddin University, Makassar 90245, Indonesia
2   Department of Agricultural Socio-Economic, Faculty of Agriculture, Hasanuddin University, Makassar 90245, Indonesia
3   Faculty of Economic, Hasanuddin University, Makassar 90245, Indonesia
*   Correspondence: darsalman@agri.unhas.ac.id

**Abstract:** A strategic issue in national development is the integration between national and regional planning. This research had the aims to: (1) analyse the consistency and integration between national and regional plans, (2) analyse the consistency of national and regional planning with the actual implementation, and (3) analyse the factors that influenced the inconsistencies of national and regional planning in the agricultural sectors. The research was carried out by the Ministry of Agriculture (MoA), Indramayu Regency, in West Java Province and South Lampung Regency in Lampung Province. The data were analysed using both content and consistency analyses and a comparative case study. The research results showed that: (1) the level of consistency between the planning documents from the MoA and those from Indramayu Regency reached 87.68% and 95.81%, when compared with those from South Lampung Regency; (2) in terms of the consistency between the documents and the actual implementation, South Lampung Regency had programs/activities which were more integrated with the MoA programs compared with Indramayu Regency; and (3) there were five main factors which caused inconsistencies: policies from the district heads and the parties involved, force majeure, the development priorities, the communication distortion, and support from the political parties. The central government (the MoA) and the Regional Agriculture Office must collaborate by making an electronic-based priority scale of development planning and by reinforcing the capacities of regional planners.

**Keywords:** consistency; planning; agriculture; centre; region

## 1. Introduction

Planning, in essence, is the application of knowledge into action to influence the direction and speed of change in reaching shared objectives [1]. The success of development planning and its implementation depends on the active involvement of the public [2–4], feedback and criticism from the public [5,6], the parties involved [7–10], aspects of trust and justice in the substance of the plan [11], the roles of the government and the public in its various stages [12], the collaboration among institutions or parties [13–15], and the series of stages in the planning process itself [16]. Therefore, the drafting process of development planning in all sectors, including the national and regional agricultural sectors, must take notice of the existing political, technocratic, and participatory aspects and the top-down and bottom-up integrations [17].

One of the main strategic issues in agricultural development in Indonesia today is the integration between national and regional planning. The existence of the Job Creation Act policy and the Circular Letter from the Indonesian Ministry of National Development Planning number 122/2020 regarding the synchronization of the planning process enables various parties and institutions to affect the technocratic process in composing plans and their implementation [18]. The involvement of various stakeholders, such as groups

or individuals, can influence or be influenced by the achievement of the objectives of a plan [19]. When connected with the Bourdieu framework [20,21], the world of planning is similar to a field in which there are a number of parties who have their own capital and habitus contest. As in the case of a game, these parties utilise their capital and specified habitus to achieve a dominant position and develop strategies to maintain that position, for the sake of their interests [22]. In this context, Freeman and McVea [23] mention that stakeholders are every group or individual who can influence or be influenced by the achievement of the objective. That is, the more powerful a stakeholder, the bigger their chance to influence the achievement of a plan's objectives in the public sphere.

Research regarding planning in the public sphere has been carried out in various countries and under various themes. D. Arsyad et al. [24] express that agricultural planning in border areas needs good institutional arrangements. F.X. Xia et al. [25] also argue that the handling of agricultural risks must be planned with the involvement of multiple parties because planning is an important part of the progress of regional economic development [26]. The appearance of the initiated and centred regional plans and the partnership process in national and regional planning will increase the effectiveness of planning [27,28] as well as the integration of national and regional planning and budgeting [29,30]. The existence of various factors which influence the occurrence of inconsistencies in planning between the central government and regional government and between plans and their implementation will also create inconsistencies between planning and its objectives [31–35]. Consistencies between national and regional planning in the agricultural sectors have not yet become a focus of the study of development planning. Studies of this aspect are important in agrarian countries, such as Indonesia, because the success of agricultural development through consistent planning between the national and regional levels will contribute significantly to the success of the development in general. It is this aspect that will be the focus of this study.

Generally, a larger region with a high enough potency of farming will require more adequate planning programs/activities and budgeting [36], as occurred in Indramayu Regency in West Java Province and South Lampung Regency in Lampung Province. These two regions are the centres of agricultural production and make large contributions to regional and national food fulfilment in Indonesia [37]. It was recorded that Indramayu Regency has an area of 54.35%, or 110,877 ha, of paddy fields in the country, producing 1,363,312 tonnes of dry milled grain or 782,132 tonnes of rice [38], and contributes to 11.95% of the needs in West Java Province [39]. Meanwhile, South Lampung Regency has an area of 40,220 ha with a harvest estimation of 217,188 tonnes [39] and a contribution of 14.16% to Lampung Province [40].

The main problems that occurred between national planning (MoA) and regional planning (Indramayu Regency and South Lampung Regency), in essence, were communication problems, in terms of verbal communication, between the planning documents and their implementation. For example, according to [41], the farmers' corporation program, which was a main program in the ministry, was not listed in the planning documents in Indramayu Regency, and agribusiness terminal activities which existed in the province and central documents, were not in the documents from South Lampung Regency, in addition to many other programs/activities that were not discrepant. These programs/activities that were not synchronous/consistent resulted in inappropriate planning with implications for the expected outputs and outcomes and an inefficient use of the budget. Therefore, the main question in this study is how consistent planning can be carried out between the central and regional governments, from the central documents to the regional documents, and from these documents to their implementation.

Research on program consistency has been conducted by [42–46]. However, this research only focuses on proving the computational geometrics theorem in a program, while research by [47], in addition to proving the theorem, also studies the consistency of the program's reasoning. Liu et al. [48] conducted research on planning consistencies and implementation in urbanizing China by comparing the urban and land use plans

in suburban Beijing by means of comparing the planning and implementation using computational maps. The authors of [49,50] conducted research related to consistency in information technology and the fast detection of unsolvable planning instances using local consistencies and information system models. All of these methods are only applicable to a computational program with a certain algorithm. Meanwhile, our research was conducted to look at programs/activities from the government, not computational programs, and thus the results of the research on computational programs are less relevant to addressing the multi-dimensional gaps in the agricultural sector.

Because of this gap, and considering the importance of the relevance between consistency and the integration of development planning in the agricultural sectors, this research aimed to assess the occurrence of consistency in planning and integration between the national planning documents to the regional planning documents and from the national and regional planning documents to the actual implementation, and identify the reasons for the inconsistencies. Through this research, it is expected that the growing gaps in the literature about planning consistency and integration in the public sphere could be filled to assist the central and regional governments and the community in the planning and implementation in the agricultural sectors.

## 2. Materials and Methods

### 2.1. Study Area

The research was carried out at the Ministry of Agriculture (MoA), Indramayu Regency, in West Java Province and South Lampung Regency in Lampung Province, as shown in Figure 1. These research locations were chosen because Indramayu and South Lampung Regencies are centres of high agricultural production and represented two different agricultural islands (Java and Sumatra islands).

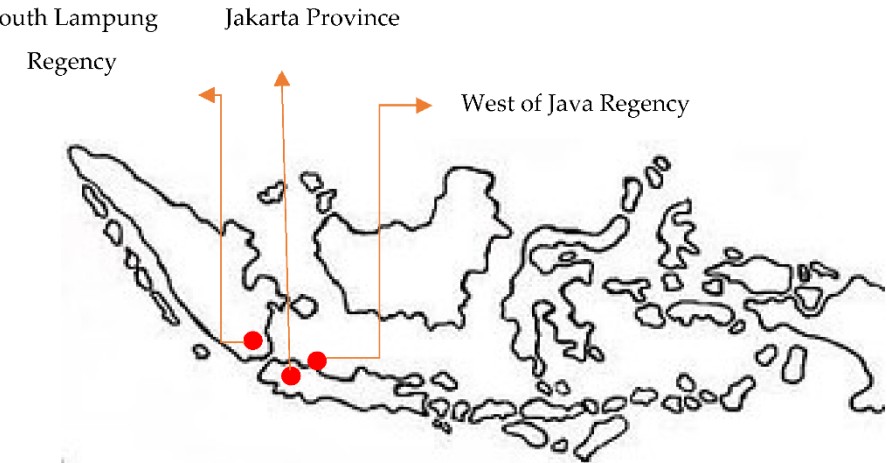

**Figure 1.** Map of Indonesia (Red Dots are Research Location Areas). Source: Taken by [51] (developed by author).

The approach used to reach the objectives of this research was through the qualitative method with case studies. The authors of [52,53] explain that the focus of case studies is specification of cases in an event which includes individuals, cultural groups or a life portrait. This implies that a case can be examined as an object of study or considered as a methodology. The case units studied were agricultural development planning programs at the national level and regional development programs, and while the focus of a case can be seen from its uniqueness, this research required an intrinsic case study.

The types of data used in this research were primary data and secondary data. The primary data used in this research were obtained through in-depth interviews with parties involved in the composition of national and regional planning, field observations, and identification of national and regional planning documents, as described in Figure 2. The

secondary data used were the planning documents from the MoA from the 2020–2024 period and the planning documents in the Regional Medium-Term Development Plan (RMTDP) and the strategic plans from the Regional Apparatus for Agricultural Affairs in Indramayu Regency and South Lampung Regency.

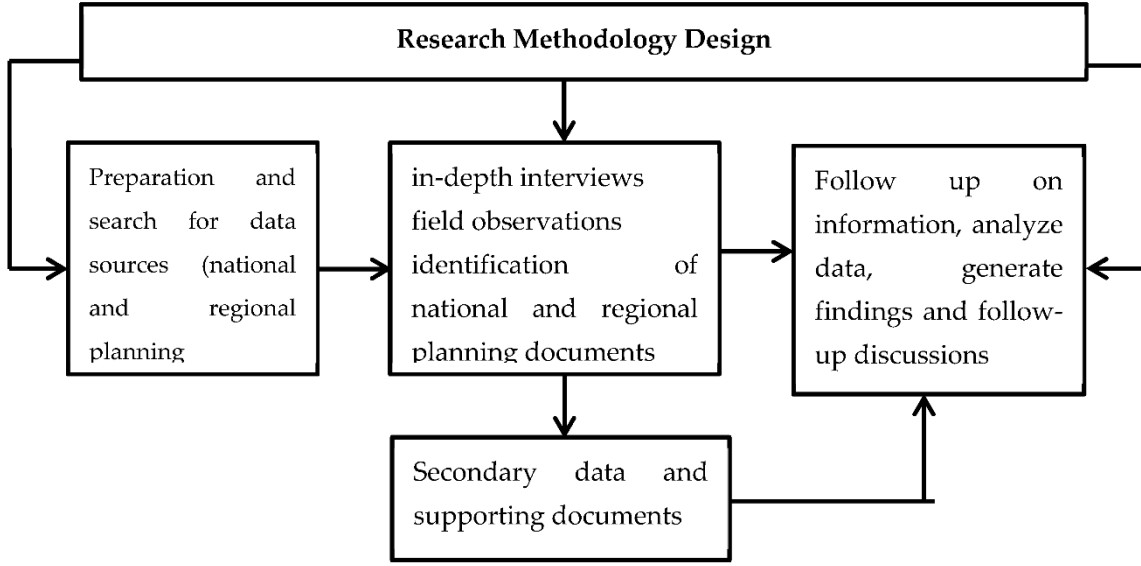

**Figure 2.** Flowchart of Data Collection Method. Source: Based on the method develop by [54].

In-depth interviews were conducted using predetermined questions and topics while allowing new topics to emerge during the interviews. as most aspects of the interview were informal, although carefully controlled [52]. The in-depth interviews were conducted with purposively selected key informants in the community and relevant government officials from the village to district, provincial, and central levels [55]. Interviews of key characters were conducted with those who influenced the planning process, which are described in Table 1.

**Table 1.** Description of Research Informants in In-depth Interviews.

| No | Informants | Number of Informants (n) |
|---|---|---|
| 1. | Expert Staff of Commission IV of HRRI in charge of the agricultural sectors | 1 |
| 2. | Planning Bureaus of the MoA | 2 |
| 3. | Planning Sections of West Java and Lampung Province | 2 |
| 4. | Planning Section of Indramayu and South Lampung Regencies | 4 |
| 5. | Regional Technical Service Unit | 1 |
| 6. | Agriculture Extensions | 2 |
| 7. | NGO | 1 |
| 8. | Farmers' Group Associations | 2 |
| Total Number of Informants | | 15 |

Source: Researcher's Analysis, 2022.

Based on Table 1, it can be seen that as many as 15 informants were selected for the in-depth interviews, namely, the key figures who influenced the planning distortion process, such as: Commission IV House of Representatives of the Republic of Indonesia (Commission IV HRRI) of the Agriculture Sectors, Planning Bureaus of the MoA, Planning Sections of West Java and Lampung Provinces, Planning Sections of Indramayu and South Lampung Regencies, a Regional Technical Service Unit, Agriculture Extension Workers, an NGO, and a Farmers' Group. Information collected in the in-depth interviews was about the consistency and integration of central planning documents and regional planning documents. Based on literature reviews, prior to conducting the main study, interviews

were conducted in which various questions were asked to encourage informants to speak freely and openly. This was also carried out in the research, where researchers asked several questions related to the planning documents and allowed informants the freedom to answer by explaining about the processes and mechanisms that had been implemented, especially in agricultural sector planning.

### 2.2. Methods of Analysis

The data analysis techniques used were content analysis and consistency analysis between plans and between plans and implementation in the development planning in agricultural sectors, specifically related to the following points.

#### 2.2.1. Content Analysis and Consistency

Content analysis is a quantitative approach that measures a text by the number of words or phrases. The measurement of frequency and intensity regarding key texts was specifically adopted by this research analysing planning and zoning codes [56]. Content analysis discussed the substance of the programs/activities in the Strategic Plans of the MoA for the period 2020–2024 and the Strategic Plans of the Regional Government Agricultural Affairs in Indramayu and South Lampung Regencies for the period 2021–2026. Furthermore, this stage examined the contents of the two planning documents starting from the formulation of the problems, goals and objectives, programs, and activities to outputs and outcomes.

The analysis of the consistency of the strategic plans was carried out on several dimensions, namely: (1) Activities in the Strategic Plans of the MoA which were in accordance with the Strategic Plans of the Department of Agriculture and implemented in Indramayu and South Lampung Regencies; (2) Activities in the regional strategic plans which were implemented in Indramayu and South Lampung Regencies but were not in accordance with the Strategic Plans of the MoA; (3) Activities in the Strategic Plans of the Ministry which were implemented in Indramayu and South Lampung Regencies but were not in accordance with the contents of the regional government agricultural affairs' strategic plans; (4) Activities implemented in Indramayu and South Lampung Regencies but were not in accordance with the contents of the Strategic Plans of neither the MoA nor the regions. The next step was compiling a table of analysis results from program contents, activities, consistencies, inconsistencies, and the percentages (%) of interrelationships between the Strategic Plans of the Ministry and those of the Regional Government Agricultural Affairs in Indramayu and South Lampung Regencies. The percentage levels of consistency were calculated using the formula:

$$\frac{\text{Consistent number of programs/activities}}{\text{Number of programs comparedkan}} \times 100\%$$

Source: [49,50].

#### 2.2.2. Analysis of the Causes of Inconsistency

A comparative case study was used to perform the analysis of the factors causing the inconsistencies between national and regional planning in the agricultural sector in Indramayu and South Lampung Districts. The comparative case study adopted what Maxwell calls process orientation. The process approach "tends to see the world in terms of people, situations, events, and processes that influence each other. Explanations are based on an analysis of how some situations and events affect others", how x plays a role in causing y and how a process links x and y [57]. Comparative case study analysis was used to compare several cases on a theme in terms of the integration of national and regional planning in agricultural sectors in Indramayu and South Lampung Regencies. An illustration of the comparative analysis process based on a case study is shown in Figure 3.

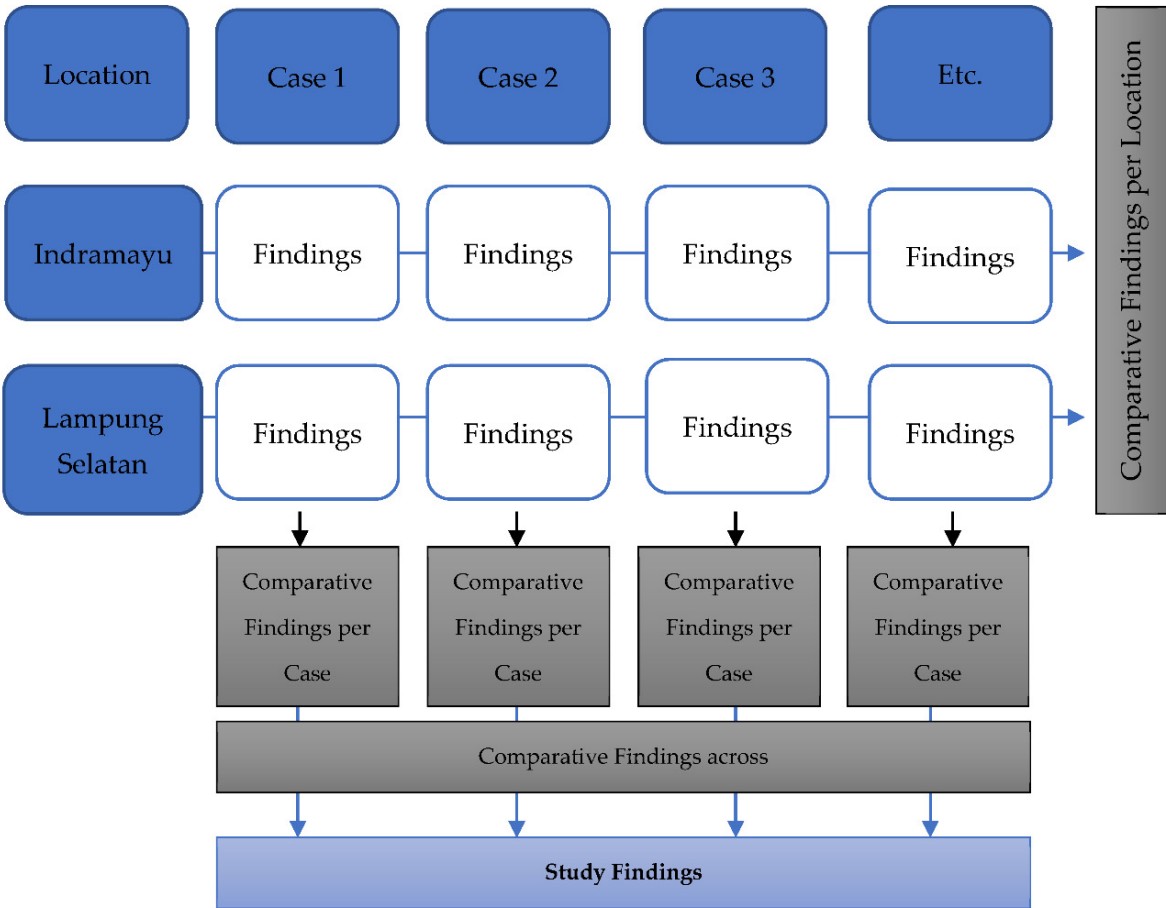

**Figure 3.** Comparative Case Study Approach. Source: Table based on discussions on comparative case study approach by [58].

## 3. Results

### 3.1. Consistency of Central Planning Documents with Regional Planning Documents in Indramayu and South Lampung Regencies

The Agriculture Office of Indramayu Regency and South Lampung Regency, in preparing their strategic plans, made several adjustments to the priority agenda of the MoA, which refers to the priority agenda of the Working Cabinet, namely NAWACITA (nawa means nine and cita means hope). For example, an adjustment was made to priority number 7: "Realizing economic independence by developing strategic sectors of the domestic economy by directing agricultural development forward to realize food sovereignty". This becomes a reference or guideline in the preparation of regional strategic plans, as contained in the background points of the Strategic Plans of the Agriculture Offices of Indramayu Regency and South Lampung Regency.

Content analysis and consistency were seen from the programs contained in the Strategic Plan documents of the MoA and the Strategic Plan documents of the Agriculture Offices of Indramayu Regency and South Lampung Regency. The five main work programs of the MoA that were used as references in this research were: (a) the Quality Food Availability, Access and Consumption Program; (b) the Industrial Value-Added and Competitiveness Program; (c) the Science and Technology Research and Innovation Program; (d) the Vocational Education and Training Program; and (e) the Management Support Program [41]. Then, the five main work programs were compared with the programs contained in the strategic plans of Indramayu Regency and South Lampung Regency.

Overall, based on the results of the analysis of the planning documents that had been carried out, Indramayu Regency was found to have a consistency level of 87.68%, while South Lampung Regency had a consistency level of 95.81% (Table 2). This means that about 12.32% of programs/activities were inconsistent in Indramayu and 4.19% were inconsistent in South Lampung. The detailed analyses of the consistency levels are shown in Tables 3 and 4.

**Table 2.** Level of Consistency between Planning Documents and Implementation in Indramayu Regency and South Lampung Regency.

| Regency | P-1 | P-2 | P-3 | P-4 | P-5 | P-6 | P-7 | P-8 | Average |
|---|---|---|---|---|---|---|---|---|---|
| Indramayu | 71.79% | 100% | 100% | 83.3% | 75% | 71.42% | 100% | 100% | 87.68% |
| South Lampung | 86.64% | 92.41% | 100% | 100% | 100% | | | | 95.81% |

Description: P: Program; Source: Analysis Results, 2022.

Table 2 shows the level of consistency between planning contained in the documents and its implementation in each program from the Agriculture Offices of Indramayu Regency and South Lampung Regency. The existence of programs in the regencies that were not in line with the national programs would create improper performance measurements and achievements of activity objectives, which would have implications for the efficiency of the budget that had been given. The details of the levels of planning consistency in the two regencies are shown in Tables 3 and 4.

Tables 3 and 4 demonstrate percentages of consistency among planning documents between central (MoA) and regional (Indramayu and South Lampung districts) governments. In Indramayu Regency, there are seven activities that are not consistent with the MoA, with a total percentage consistency of 87.68%. On the other hand, in South Lampung Regency there is one activity that is not consistent with the MoA's program. The average percentage consistency for South Lampung Regency is 95.81%.

*3.2. Consistency between Planning Documents and Implementation in the Indramayu and South Lampung Districts*

At this stage, the four analyses were performed on the contents in the activity plans to assess the consistency of the activities. These analyses were: (1) the activities implemented in Indramayu/Lampung Selatan that were actually in line with those proposed in the central government's planning documents (Table 5); (2) the activities implemented in Indramayu/Lampung Selatan that were not in line with those proposed in the central government's plans (Table 6); (3) the activities implemented in Indramayu/Lampung Selatan that were not in line with those in the local government planning documents (Table 7); and (4) the activities implemented that were not in line with what is stated in both the central and local government planning documents.

Table 3. Detailed Level of Consistency of Planning Documents between MoA and Indramayu Regency.

| MoA Activities | Indramayu Activities | C | NC | Percentage of Consistency |
|---|---|---|---|---|
| Program 1. Quality Food Availability, Access and Consumption Program (Cr, Ho, Fa, Afi, Fs, and Qr)Activities: | Program 1. Increase Production and Quality of Food Crops | | | 71.79% |
| 1. Development of rice, corn, soybeans, local foods | Activities: | C | | 33.3% (except corn and local foods) |
| 2. Cr, Ho, and livestock seed/nursery | 1. Increase Production of Rice Through Jajar Legowo Planting System | | | |
| 3. Increase production of vegetables and medicinal plants | 2. Rural Agribusiness Development (RAD) Assistance | C | NC | 0% |
| 4. Increase production of fruits and floriculture | 3. Agricultural Business Development and Empowerment | C | | 100% |
| 5. Agricultural area development to support Food Estate (FE) and corporations | 4. Handling of Harvest and Post-Harvest as well as Observation of Agricultural Yield Loss | C | | 100% |
| 6. Handling of plant pests | 5. Subsidized Fertilizer Escort | C | | 100% |
| 7. Provision of feed and increase production of livestock | 6. Food Crop Protection | C | | 100% |
| 8. Provision of feed and increase production of livestock | 7. Development of Vegetable Pesticides and Organic Agriculture of Food Crops | C | | 100% |
| 9. Animal disease control | 8. Promotion of Agricultural Products | | NC | 0% |
| 10. Increase in Veterinary Public Health | 9. Preparation of Food Production Database | C | | 100% |
| 11. Irrigation water management, land expansion and protection | 10. Development of Mung Beans Planting | | NC | 0% |
| 12. Provision of pre-harvest agricultural tools and machinery, facilitation of fertilizers and pesticides, agricultural financing | 11. Development of Soybeans Planting | C | | 100% |
| 13. Food diversification | 12. Development of Soybean Seed Breeding | C | | 100% |
| 14. Implementation of quarantine | 13. Procurement of Farmer Cards for Insurance Participants | C | | 100% |
| Program 2. Industrial Value Added and Competitiveness Program (Cr, Ho, Pl, and AF)Activities: | Program 2. Disaster Early Warning | | | 100% |
| 1. Provision of post-harvest agricultural tools and machinery and processing of agricultural products | Activities: 1. Facilitation of Climate Team | C | | 100% |
| 2. Processing and marketing of agricultural products | Program 3. Data/Information Development | | | 100% |
| 3. Development of plantation seeds | Activities: 1. Preparation of Food Production Database | C | | 100% |
| 4. Development of plantation areas based on smallholder corporations | Program 4. Capacity Building of Agricultural Extension | C | | 83.3% |
| 5. Provision of infrastructure to support agricultural corporations | 1. Performance Assessment of Agricultural Extension | C | | 100% |
| 6. Support for thematic activities (new renewable energy, frontier/underdeveloped) | 2. Agricultural Extension Training | C | | 100% |
| Program 3. Science and Technology Research and Innovation Program | 3. Improvement of Agricultural Extension Facilities | C | | 100% |
| 1. Assembling superior varieties of Cr, Ho | 4. Implementation of Applied Local Specific Technology Studies/Appropriate Pilots | C | | 100% |
| 2. Assembling cultivation technology | 5. Agricultural Extension | C | | 100% |
| 3. Propagation of seed sources resulting from agricultural research and development for innovation | 6. Development of Rural Extension Program | | NC | 0% |
| 4. Downstream agricultural technology innovation | Program 5. Farmer Institutional Improvement | | | 75% |
| Program 4. Vocational Education and Training Program | Activities: 1. Training of Farmers (Empowerment of farmer groups) | C | | 100% |
| 1. Providing BOP incentives for extension workers | | | | |
| 2. Pilot application of agricultural technology | 2. Organizing Workshop Pulpits | | NC | 0% (Regent's discretionary activity) |
| 3. Implementation of counselling | | | | |
| 4. Counselling escort in 34 provinces | 3. Capacity Building of Farmers' Institutions | C | | 100% |
| 5. Escorting farmer corporations at food production centres | | | | |
| 6. Vocational training and professional certification in agriculture | | | | |
| 7. Implementation of technical guidance for farmers and extension workers | | | | |
| 8. SARTC development and strengthening | | | | |
| 9. Development of young agricultural entrepreneurs (YAEN) | | | | |
| 10. Implementation of agricultural vocational education | | | | |

**Table 3.** *Cont.*

| MoA Activities | Indramayu Activities | Analysis of Consistency | | Percentage of Consistency |
|---|---|---|---|---|
| | | C | NC | |
| | 4.   Preparation of subsidized fertilizer | C | | 100% |
| | Program 6. Improving Horticultural Production and Quality | | | 71.42% |
| | 1.   Development of Gedong Gincu Area | | NC | 0% (regional priority activity) |
| | 2.   Development of Vegetable Areas | C | | 100% |
| | 3.   Horticulture to support Regional Achievements | | NC | 0% (Regent's discretionary activity) |
| Program 5. Management Support Program (Binding Expenditures, Capital, Operations, Digitization, Policy Analysis, Supervision, Operational Extension Workers, etc.) (All Echelon I Work Units) | 4.   Increase Promotion of Horticultural Products | C | | 100% |
| 1.   Preparation of planning and evaluation | 5.   Development of Horticultural Business | C | | 100% |
| 2.   Coordination and cooperation abroad | 6.   Development of Horticultural Production | C | | 100% |
| 3.   Improvement of public services and procurement | 7.   Horticultural Crop Protection | C | | 100% |
| 4.   Improvement of public relations services and public information | Program 7. Enhancement of Production and Quality of Plantation Plants | | | 100% |
| 5.   Development of agricultural data and information | Activities: | | | |
| 6.   Protection of plant varieties | 1.   Development of People's sugarcane. | C | | 100% |
| 7.   Implementation of SPIP performance audit and escort services | 2.   demonstration plot of Sugarcane Plants | C | | 100% |
| 8.   Office operations (salary, honorarium, capital expenditure, etc.) | 3.   Sugarcane Nursery | C | | 100% |
| 9.   Agricultural digitization | 4.   Rehabilitation of Coconut Plants | C | | 100% |
| 10.   Program/activity supervision | 5.   Optimization and Enhancement of Plantation Commodity Production | C | | 100% |
| 11.   Operational costs of extension workers | 6.   Construction of Plantation Commodity Production Supporting Infrastructure | C | | 100% |
| 12.   Implementation of secretarial services for Es.1 of MoA | 7.   Prevention and Control of Plantation Pests and Diseases | C | | 100% |
| | Programs 8. Empowerment of Agricultural Resources | | | 100% |
| | 1.   Construction/Rehabilitation of water network, water sources and production roads | C | | 100% |
| | 2.   Development of Agricultural Machinery | C | | 100% |
| | 3.   Optimizing the Utilization of Tools and Machinery | C | | 100% |
| | 4.   Development of Food Crops Facilities and Infrastructure | C | | 100% |
| | 5.   Development of Horticultural Facilities and Infrastructure | C | | 100% |
| | 6.   Construction of Plantation Commodity Production Supporting Infrastructure. | C | | 100% |
| | Average % of consistency | | | 87.68% |

Description: C (Consistent); NC (Not consistent); Cr: Crops; Ho: Horticulture; Af: Animal Farm; Afi: Agricultural Facilities and Infrastructure; Fs: Food Security; Cr: Quarantine; Pl: Plantation; SARTC: Self-help agricultural and rural training centre; Source: Analysis Results, 2022.

**Table 4.** Detailed Level of Consistency of MoA Planning Documents with South Lampung Regency.

| MoA Activities | South Lampung Activities | Analysis Consistency | | Percentage Consistency |
|---|---|---|---|---|
| | | Consistent (C) | Not Consistent (NC) | |
| | Programs 1. Enhancement Production | | | 86.64% |
| Program 1. Availability, Access, and Consumption Program Food QualifiedActivities:<br>1. Development of rice, corn, soybeans, local food<br>2. Cr, Ho, and livestock seed/nursery<br>3. Enhance production of vegetables and medicinal plants<br>4. Enhance production of fruit and floriculture<br>5. Development in the area of agriculture support FE and corporations<br>6. Handling organism bully plants<br>7. Preparation of feed as well as enhancement of production cattle<br>8. Preparation of feed as well as enhancement of production cattle<br>9. Control and countermeasures of animal disease<br>10. Enhancement Veterinary Public Health<br>11. Irrigation water management, expansion, and protection of land<br>12. Preparation of Alsintan pre- harvest, facilitation of fertilizers and pesticides, financing agriculture<br>13. Diversification of food<br>14. Maintenance of quarantine<br><br>Program 2. Value-Added and Power Program Competition IndustryActivities:<br><br>1. Preparation of Alsintan post-harvesting and processing results in agriculture<br>2. Processing and marketing results in agriculture<br>3. Development of seed plantation<br>4. Development of plantation-based corporation farmers<br>5. Preparation of means of infrastructure support for corporation agriculture<br>6. Support of thematic activity (energy renewable, frontier/lagging)<br><br>Program 3. Research and Innovation of Program Knowledge Knowledge and Technology<br><br>1. Assembly of superior Cr, Ho varieties<br>2. Assembly of cultivation technology<br>3. Propagation of seed source results and innovation in agricultural research and development<br>4. Downstream innovation in agricultural technology | 1. Production of superior commodity plant food (rice and corn) | C | | 66.60% (there is a focus on development of commodity soy and not yet a focus of activities developing the farming sector farm) |
| | 2. Production of superior commodity plant horticulture (big chili and red onions) and development of agrotourism | C | | 66.60% |
| | 3. Production of commodity coconut | C | | 100% |
| | 4. Production of commodity kako | C | | 100% |
| | 5. Production of commodity coconut palm | C | | 100% |
| | Program 2. Provision and Development of Agricultural Facilities | | | 92.41% |
| | 1. Supervising the Use of Supporting Facilities in Agriculture in accordance with Commodity, Technology, and Location-Specific objectives | C | | 100% |
| | 2. Accompaniment use means supporter agriculture | C | | 66.60% (there is no activity in accompaniment facilitation or financing agriculture) |
| | 3. Guarantee Purity and Sustainability of Animals/Plants | C | | 100% |
| | 4. Coordination and Synchronizing Infrastructure Supporter other | C | | 100% |
| | 5. Construction, Rehabilitation, and Maintenance of Moderate Groundwater Irrigation | C | | 100% |
| | 6. Construction, Rehabilitation, and Maintenance of dam Agriculture | C | | 100% |
| | 7. Construction, Rehabilitation, and Maintenance of Farm Roads | C | | 100% |
| | 8. Construction, Rehabilitation, and Maintenance of Trench DAMs | C | | 100% |
| | 9. Construction, Rehabilitation, and Maintenance of Long Storage | C | | 100% |
| | 10. Construction, Rehabilitation, and Maintenance of Sluice | C | | 100% |
| | 11. Construction, Rehabilitation, and Maintenance Hall of the Counsellor in the District as well as means supporters | | NC | (50%) Hall counseling should enter the program |
| | Program 3. Agricultural Business Licensing Program | | | 100% |
| | 1. Activities:Coaching and supervision of application for agricultural permissions | C | | 100% |
| | Program 4. Control And Countermeasures Disaster Agriculture | | | 100% |
| | 1. Countermeasures post-disaster in natural fields, plant food, horticulture, plantations, etc. | C | | 100% |
| | 2. Enhancement of Institutional Capacity of Agriculture Counsellors in Districts and Villages | C | | 100% |

**Table 4.** *Cont.*

| MoA Activities | South Lampung Activities | Analysis Consistency | | Percentage Consistency |
|---|---|---|---|---|
| | | Consistent (C) | Not Consistent (NC) | |
| | Program 5. Extension of Agriculture Programs | | | 100% |
| | 1. Development of Institutional Capacity of Farmers in the Districts and Villages | C | | 100% |
| | 2. Provision and Utilization of Facilities and Infrastructure Counselling in Agriculture | C | | 100% |
| | 3. Dissemination of technical, social, economic, and innovative information about agriculture | C | | 100% |
| | 4. Implementation of farmer counselling and empowerment | C | | 100% |
| | 5. Formation and strengthening of institutional farmer corporations | C | | 100% |
| | 6. Accompaniment management corporation farmer | C | | 100% |
| | 7. Evaluation of the feasibility and standardization of farmer corporation management | C | | 100% |
| | % Average Consistency | | | 95.81% |

Description: C (Consistent); NC (Not consistent); Source: Analysis Results, 2022.

**Table 5.** Activities implemented in Indramayu/Lampung Selatan and appropriate with Central Document.

| Research Location | Activity | Statement |
|---|---|---|
| Department of Agriculture Regency Indramayu | P1 = 12 activities<br>P2 = 1 activity<br>P3 = 1 activity<br>P4 = 6 activities<br>P5 = 3 activities<br>P6 = 5 activities<br>P7 = 7 activities<br>P8 = 6 activities<br>Total 41 Activities | "Most activities are appropriate, just the allocated areas for mangoes, green beans, other supporting horticultural achievements and the discussion forum are different"<br>(IS, P, 48, Planning Division of the Department of Agriculture Indramayu Regency) |
| South Lampung Regency | P1 = 4 activities<br>P2 = 11 activities<br>P3 = 1 activity<br>P4 = 2 activities<br>P5 = 7 activities<br>Total Activities: 25 | "A lot of the regency activities are relevant, but those of agro-tourism were specifically requested by the head of the regentdistrict himself and the location had been designated at the back of the office"<br>(HN, Functional Agricultural Service Planner South Lampung Regency) |

Description: P = Program.

**Table 6.** Activities implemented in Indramayu/South Lampung that are not in line with the documents of the central government.

| Research Location | Activity | Statement |
|---|---|---|
| Department of Agriculture Regency Indramayu | (1) Area development of Gedong Gincu | "Highlighted regional activity funded by the West Java Province budget with a program allocation in Jatibarang District, as an effort to develop the regional potential of manggoe fruit in Indramayu regency"<br>(IS, P, 48, Planning Agency of the Agriculture Regional Office of Indramayu Regency) |
| | (2) Horticulture supporting regional achievement | "That locally initiated activity was in the form of donating plant seedlings for the health service units at schools in the preparation for Indramayu participation in the healthy school competitions or health community centres competitions in the province."<br>(IF, L, 36, Planning Agency of the Agriculture Regional Office of Indramayu Regency) |
| | (3) Planting Development of Green Beans | "This activity of commodity development of green beans emerged from the regional strategic plans particularly from the Regent's strategic plans as a support of the stanting program, covering the area of 100 hectares"<br>(IF, L, 36, Planning Agency of the Agriculture Regional Office of Indramayu) |
| | (4) Discussion Forum | "This activity aiming at increasing the farmer capacity building comes from the level 2 of Region Budget and Revenues Plans (BRPII)"<br>(IF, L, 36, Planning Division of the Agriculture Regional Office of Indramayu Regency) |

**Table 6.** *Cont.*

| Research Location | Activity | Statement |
|---|---|---|
| Department of Agriculture South Lampung Regency | (1) Horticulture Agrotourism Area Development | "This activity is the highly prioritized activity of the elected Regent, and is stated in the Strategic Plans of the Lampung Selatan Regency. However, it is not in either the plans of the province or the plans of the Agriculture Minstry. It is partly as a form to support the programs of the Regent." (HN, An official in Planning Division of Agriculture Regional Office of Lampung Selatan Regency) |

Source: Analysis Results, 2022.

**Table 7.** The activities implemented in Indramayu/Lampung Selatan regencies not in line with the regency development plan documents.

| Research Location | Activity | Statement |
|---|---|---|
| Department of Agriculture Regency Indramayu | Milennial Farmers | "The activity was designed when the process of the strategic plans was focusing only on increasing the agriculture products. Looking at the development of the situation and the current issues, it was decided that this activity was required to do initially, and therefore, it was included in the following strategic development plans of the region." (IF, L, 36, Planning Division of the Agriculture Regional Office in Indramayu Regency) |
| | Farming insurance | |
| | Farmers' Corporation | "Starting from a joint of farmer groups, it has become a corporation that can offer solutions to farmers and can play a role in the social and economy development of the community. This is the first corporation established in Indramayu and has been coached and mentored by Bank Indonesia (BI). In 2019 it was decided to become a corporation by the central Government, with the areas of District Cikedung, Terisi, dan Lelea." (IS, P, 48 Planning Division of the Agriculture Regional Office of Indramayu Regency) |
| Department of Agriculture South Lampung Regency | Competitive Agrobusiness and Agroindustry | "This activity item comes from the Province Budget of Lampung Porvince, but actually it was not in any of the strategic plans of ours here in the region" (HN, An official in Planning Division of the Agriculture Regional Office of Lampung Selatan Regency) "This activity plan was made and implemented by the previous Governor and now it is not of our priority any longer.i" (RZ, Agriculture Agency of Lampung Provice) |
| | Area Development of the Agribusiness Terminals | "This activity was carried out with the budget from the Lampung Province Budget, but now the regent's program policies are directed to focus on improving the agriculture products" (HN, P, An official in the Planning Division of the Agriculture Regional Office of Lampung Selatan) "As there is no budget available for that activity now, it is only relying on that of the Province budge." (SJ, Mentoring Division of the Agriculture Regional Office of Lampung Selatan) |

Source: Analysis Results, 2022.

### 3.2.1. Activities Implemented in Indramayu/Lampung Selatan Regency and their Consistencies with the National Plan Documents

It is considered to be important to have consistency in planned activities between the central and the local governments and in the implementations of each. However, local governments do have the authority to have their own activities/special activities separate from those of the central government. This is due to several factors. Table 5 describes the

number of implemented regency activities (Indramayu and Lampung Selatan) that were in line with the central government activities (Agriculture Ministry).

The results of the analysis show that there were eight programs with 41 total activities implemented in Indramayu Regency that were consistent with those of the central government. Meanwhile, in Lampung Selatan Regency, there were five programs with 25 total activities that were consistent with those of the central government. A few inconsistencies between the activities of the regencies and the central government can be brought up by a number of factors. As stated by [59], local governments can actually make their own policies that are not against the policies of the central government in their efforts to explore their own regional potentials. However, it is recommended that the programs/activities are in accordance with the national agenda so they are still set to achieve the national development goals.

3.2.2. Activities Implemented in Indramayu/South Lampung That Are Not in Line with the Documents of the Central Government

Strategic plan documents are references for the short-term goals of a nation [60], which have an orientation to achieving the expected results in 1–5 years. However, in this process, there are some additional incidental regency activities that differ from those of the central government that had to be executed, as shown in Table 6.

The results of this analysis show that there were a number of activities in the strategic plans of regional governments that had been implemented which were not stipulated in the national strategic development plans. Those activities were commonly planned and executed with the intention to support the policies made by the elected regents. The regency of Indramayu has four activities specifically designed to align with the elected regent's policies and also to develop the existing agriculture potentials that the regency has. The four activities were: (1) the local development of mango plantation in Gedong Gincu; (2) horticulture supporting regional achievement; (3) the development of a green bean planting program; (4) and the implementation of a discussion forum. Meanwhile, in the regency of Lampung Selatan, there was one activity carried out that had not been stipulated in the national strategic development plans, that is, the development of horticultural agrotourism, which was a particular activity initiated by the elected regent there.

3.2.3. The Activities Implemented in Indramayu/Lampung Selatan Regencies Not in Line with the Regency's Development Plan Documents

There have been some central government activities implemented in the regencies that were not stipulated in their regional strategic development plans. This can possibly happen due to the consideration of the strategic issues coming up, the community's interests in agricultural development programs, and the external support in the process of the regional programs, as shown in Table 7.

The analysis results show that there are four activities implemented with no reference to the regional planning documents, a few of which are: (1) activities to develop young entrepreneurship in millennial farmers (aged 18–35 years) through the activity of Developing Young Farmer Entrepreneurs (DYFE); (2) farming activity insurance for rice farmers through the Insurance for Rice Farmer Activity program (IRFA), which would provide protection against lost harvests; and (3) the development of farmer units through farmer corporations. Meanwhile, in Lampung Selatan Regency, there were two such activities, which are: (1) a competitive program for agrobusiness and agroindustry and (2) the development of the agrobusiness terminal area. Those two activities are parts of the strategic programs of the Directorate of Processing and Marketing Agriculture Products of the Agriculture Ministry which were implemented in Lampung Selatan Regency but which were not receiving enough support, either in monitoring or financing, as they were not stipulated in the documents of the regional plans.

### 3.2.4. Activities Implemented in Indramayu/South Lampung Not in Line with What Is in Either the Regional or National Planning Documents

Activities that are implemented in a region but are not in the region's planning documents are commonly those implemented by international agencies or Non-Governmental Organizations (NGOs). However, in both Indramayu and Lampung Selatan regencies, all international agency activities have been stated in both the regional and national activity planning documents. In Indramayu Regency, for example, there are two activities from an international agency aiming to modernize water irrigation and rehabilitation: the Strategic Irrigation Modernization and Urgent Rehabilitation Project (SIMURP) and the Integrated Participatory Development and Management of Irrigation Program (IPDMIP). The management of these programs is under the four ministries and one national agency, which are the National Planning and Development Board, the MoA, the Ministry of Public Work and Housing, and the Ministry of State Affairs:

> *"We have the donated activities of SIMURP and IPDMIP in a some districts, the activities ran well and we had the monitoring activity fund as well, so it goes to the activity plan documents in the Agriculture Regional Agency,"*
>
> *(IF, L, 36, Planning Division of the Agriculture Regional Office of Indramayu Regency)*

Based on the analysis results above, the level of consistency between the activities implemented in those two research locations is shown in the Venn diagram in Figure 4.

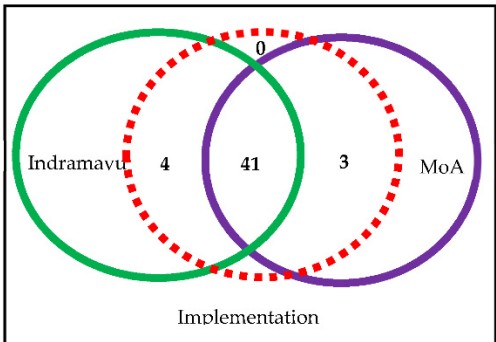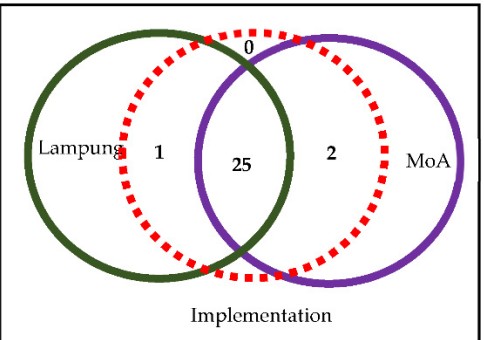

**Figure 4.** Venn Diagram showing Consistency of National Activities and Regional Activities. (Left: Indramayu Regency; Right: South Lampung Regency). Source: Analysis Results, 2022.

### 3.3. Factors That Influence the Inconsistency between National Plans and Regional Plans in the Agricultural Sector

The consistency between the documents of the planned activities of the Agriculture Ministry and those of the regions is important to highlight. This is mainly because it has become one of the indicators of the local government performance and has a lot to do with the achievement of the vision, mission, goals, and policies that have been planned [61]. Consistency in planning has a crucial role in carrying out regional development activities aiming to improve the welfare of the regional community, although, in practice, there are some factors hindering the planning between the central and local activities consistent. The level of consistency in the regional and national agriculture plans is notably evident.

Based on the similarities and the differences in the cases from the of the comparative case studies, five influencing factors are identified to be affecting the inconsistencies between national strategic plans and the strategic plans in the agricultural sector in the Indramayu Regency and Lampung Selatan Regency. They are: (1) policies of the regencies heads and those of a third-party actor; (2) force majeure; (3) agriculture product priorities; (4) communication distortion; and (5) political party interventions (Figure 5).

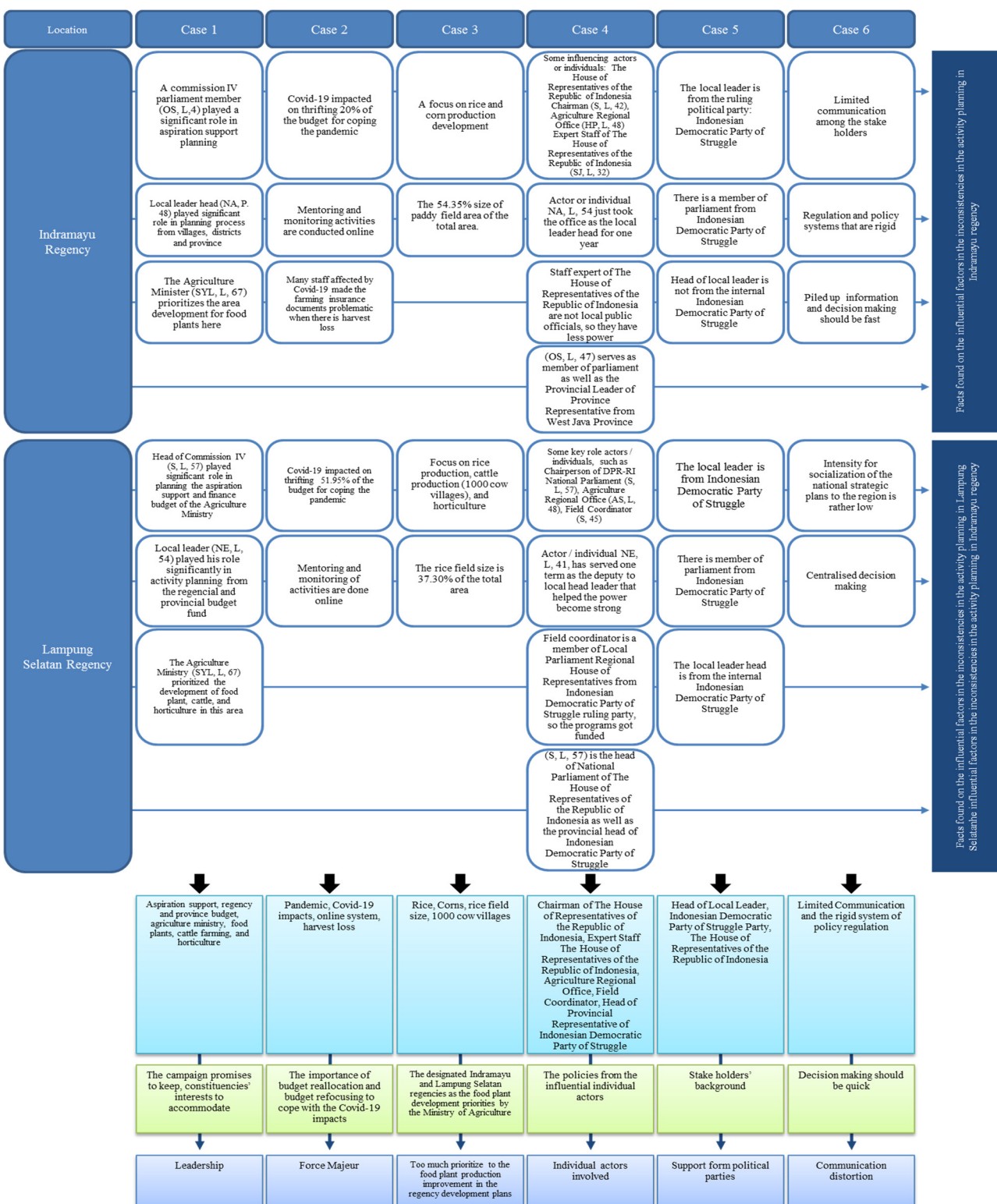

**Figure 5.** Comparative case study. Source: Analysis Results, 2022.

## 4. Discussion

In the agricultural sector, the level of consistency between the national plans and the regional ones in the regencies of Indramayu and Lampung Selatan, as well as between their planning documents and their actual implementation, is evidently high. The percentage of document consistency between the ones in the Agriculture Ministry and the ones in the

regency of Indramayu reaches 83.3% while that of Lampung Selatan is 88.8%. From the perspective of the consistency in the activity plan documents of the Agriculture Ministry and the activities implemented in the regencies, Lampung Selatan has more integrated programs/activities in common with the Ministry than Indramayu. Such high or low levels of consistency and integration between the plans made at the national level and the plans made in the regions relies on a number of factors. In Indramayu, four main factors were found to be affecting the inconsistency in the documents there, which are the leadership of the regent as the leader, force majeure, development priorities, and the involvement of specific individuals. Quite similarly, in Lampung Selatan, the affecting factors found are: the policies of the regent, force majeure, development priorities, and the involvement of specific individuals, plus the dominant political support of the ruling parties that also affects regional planning decisions.

First: The Factor of the Policies of Local Leadership. Leadership in the local government becomes one of the main factors deciding what in included in regional planning documents. The contents of strategic planning documents are influenced by the visions, missions, and programs of the local leaders. The event of a change of leadership among the elected local leaders can have an impact on programs that require some adjustment based on the policies of the elected leaders. This is considered appropriate with the stated main tasks and functions of the regional agricultural office, which are to support the activities set by the regents referring to the scope of the office. The information on this topic received from the insiders IKH, IS, and CR is presented below:

*"The recent policies of the now elected Regent are to develop the local agriculture potentials, such as gedong gincu manggoes and agro areas, thus, the planning and budgeting are directed and allocated towards implementing those two programs."*

*(IKH, Planning Division of the Agriculture Regional Office of Indramayu Regency)*

*The individuals or agencies who play some roles in the process of making the strategic plans are universities, NGOs, agriculture activists (leaders of the Farmers and Fishermen Group), farmers, Public Works and Housing Ministry, Regional Offices in regencies, farmer coaches, the regional development planning board."*

*(IS, Planning Division of the Regional Agriculture Office of Indramayu Regency)*

*"Such involvement from those individuals or agencies is thought important as that can provide more effective contributions on the ideas dan the activities. However, this is where sometimes our ideas can be different from the plans set in the above levels"*

*(CR, NGO activist in Indramayu Regency)*

Similar to the conditions in Indramayu, the policies made by the regent of Lampung Selatan also play a role in creating inconsistencies in activity/program planning. A number of activities/programs that have previously been decided as not being in line with the national agenda were eventually included in other programs in order to accommodate them in the planning and budgeting:

*"The regent has his own programs during his election campaigns and wishes to enhance the regional potential, this has made the planning become different, which eventually would accommodate the those programs in other programs, thus, they are actually rather in line"*

*(Ftr, Official in Planning Division of Agriculture Regional Office of Lampung Selatan)*

Based on the results of the conducted analyses, the findings of this research are relevant to the findings of [62,63] stating that, in the planning process, the factor of local leadership has a strong influence on the planning and actual implementation of programs/activities. A similar notion was also noted in the research of [64], which highlighted that in budget planning, the involvement of specific individuals with some political influence can possibly cause some inconsistencies. In this research, the agricultural development plans carried out in the Indramayu Regency and Lampung Selatan Regency were also influenced by

some individuals with personal interests and with influence on the policies of the regents. This eventually affected the consistency between the national plans and the regional ones in these regencies. However, one distinctive finding is that in Indramayu and Lampung Selatan, the activities that were not in the national plan were later merged and included in other activities to accommodate them in the planning and budgeting system, such as the Gedong Gincu Mango Area program in Indramayu, which has been merged into the Vocation Training and Education program, or the activity of the Area Development of Edu-tourism in Lampung Selatan, which has been combined with the Availability, Access, and Consumption of Quality Food Program.

Secondly: The Force Majeure Factor. The specific emergency situation (force majeure) referred to here was the COVID-19 pandemic that happened all over the world [65,66]. The COVID-19 pandemic has caused countries to apply measures such contact tracing, physical distancing, increased hand hygiene, face masks, and selective isolation and quarantine, followed by lockdown policies [67–69]. This has resulted in the shifting of regional budget plans among organization units and among activities/programs in terms of the kinds of spending budgets [70]. During the COVID-19 pandemic, the national government prepared several strategies to cope with that the virus, such as increasing the budget for healthcare, social donation packages, support for economic recovery, and also new policies on regional finances [71]. In order to ensure budget availability, the government emphasized the importance of the reallocation and refocusing of the public funds to face the impacts of the COVID-19 pandemic. This hindered the implementation of the programs of the Agriculture Regional Office of Indramayu and Lampung Selatan, as commented by IKH and HN:

> *"here, the budget has been used, almost all staff members were infected by the virus and the budget was realocated to handle the impacts of COVID-19"*
>
> *(IKH, Planning Division of Indramayu Regency)*
>
> *"The Agriculture Regional Office of Indramayu experienced refunctioning of its budget as much as 20% to deal with those impacted by the COVID-19"*
>
> *(HN, Official of Planning Division of Lampung Selatan Regency)*

Another research work by [72,73] concluded that fiscal strategy policies in the implementation of the national agenda are also influenced by spending in urgent conditions. The measures mention previous were taken to handle the urgent condition of COVID-19. Such measures of reallocating funds from other budgets to addressing the pandemic were also taken by developed countries. In our study on the factors hindering consistency in the strategic plans of the Indramayu and Lampung Selatan regencies shows that the agriculture sector had to experience a budget cut to its programs, as said by IS:

> *"The budget of the regional office had to be cut due to the pandemic, so several programs had to stop in adjusting to the available budget.t."*
>
> *(IS, Planning Division of Agriculture Regional Office)*

Third: The Factor of Prioritizing Agricultural Production. Programs that have been arranged in strategic area plans still focus on the enhancement of agricultural production. This occurs due to limitations in the regional income generated by the Indramayu Regency and the South Lampung Regency. Due to that fact, programs related to latest issues are yet to be accommodated. In addition, the millennial farmers program, food crop insurance, and the farmer corporation in Indramayu Regency have yet to receive support from the regional budget. This study contradicts the findings of [74], which states that the reason for inconsistent planning towards performance targets is due to the low quality of office planners and low commitment to maintaining consistency in planning. In Indramayu and South Lampung regencies, the factor of limited regional income becomes the main reason for such inconsistency:

> *"Many central and provincial programs are allocated in the regions and we cannot just carry out monitoring because of unavailable budge, the average figures are done and or in physical forms, meanwhile assistance budget is not even allocated."*

*(FTR, Planning Division of Agriculture Service South Lampung Regency)*

*"The regent puts many hopes on the rice of agriculture production, so that all this time we have been focusing on the achievements of number production, especially on commodities that are already commonly plant in this area."*

*(HS, Head of UPTD Indramayu)*

Fourth: Factor of Distorted Communication Between the Local and Central Governments. In general, there are five types of distortion, namely, market distortion, cost distortion, cognitive distortion, art distortion, and communication distortion. Communication distortion is a change in the meaning of some information/messages, either on purpose or not on purpose, that results in a change in the delivered information. It can also be defined a less accurate or a different meaning of messages/information sent in a communication process. This distortion in communication can be verbal (speech) and non-verbal (documents). The research results show that the communication between the national government and the region of Indramayu Regency has been considerably less intensive compared with the communication with the South Lampung Regency. This can be seen from the available documents on the rice farmer corporation program in Indramayu Regency and in the livestock corporation program in South Lampung Regency. These programs were not found in any regional planning, yet they both have become the priority programs of the MoA.

The process of designing regional planning must be conducted collaboratively, arranged in such a way that there is consistency in and integration of policies from both the national government, which is, in this case, represented by the Ministry of Agriculture, and the regional government. Because, basically, the development planning in the sector of agriculture is directed towards improving the welfare of farmers [75], it is expected that the programs should be in sync with that goal. Furthermore, the policies can be less top-down in nature and can be more comprehensively strategic, as expressed by HN:

*"There is still a lack of socialization and less program integration from the central government about the strategic plans from the Ministry of Agriculture, due to the fact that a lot of things only refer to the Ministry of State Affairs."*

*(HN, Official in Planning Division of the Agricultural Regional Office of South Lampung Regency)*

*"So far the plans designed have involved the national legislative members from the region represented; we would know when it is half way in the process, so we would just have to accept them and go with those proposed plans as the programs have taken place."*

*(SJ, Expert Staff for Commission IV of the National Legislative Member—HRRI—representing the Indramayu regency)*

Regional development planning, according to [76,77], are defined as a drafting process of the activity stages that involve various elements. One of which is the existence of integration of programs or activities between the central and regional governments. The factor causing inconsistencies in this integration is identified as a communication distortion, as communication is the basic requirement in establishing coherence in activity implementation.

Fifth: Factor of Supporting Party Politics. One of determining factors in planning and policy making is the political factor [60]. The power, control, and relations of individual actors in South Lampung made this regency area have more benefits than Indramayu. The fact that no such political actors operate in Indramayu reflects the imbalances in the implementation of the policies found there caused by their absence. On the other hand, the role of the legislative members in the central government representing the region of Indramayu have not been as evident as the roles played by those representing South Lampung Regency, one of which happened to serve as the chairman of the Commission IV of the National Legislative Body—HRRI. Thus, this individual has a lot to do with the consistency and integration between programs that are planned in the central government (MoA) and those that are planned in the regional government (Indramayu Regency), as the

Commission IV HRRI partners directly with the Ministry of Agriculture. So, the existence of the power of individual actors, as mentioned before, gives South Lampung a particular privilege in ensuring consistency in the planning process and implementation between the national and in the regional contexts.

## 5. Conclusions

This study aimed to analyse the consistency and integration between national and regional plans, also including the factors that may influence the inconsistencies of national and regional planning on the agricultural sector. Based on the Strategical Planning of the MoA 2020–2024, the Strategical Planning of Indramayu Regency's Department of Agriculture 2016–2021, and the Strategical Planning of South Lampung Regency's Department of Agriculture 2021–2024, the study found that are both consistent and non-consistent programs between the planning documents and the implementations: (1) The level of consistency amongst the planning documents of the MoA and those of Indramayu Regency reached 87,68% and reached 95,81% in South Lampung. (2) There were four activities that were implemented in Indramayu Regency that no longer exist in the central government's documents and three activities that no longer exist in the regional government's documents. On the other hand, there is one activity that was implemented in South Lampung Regency that no longer exists in the central government's documents and two activities that no longer exist in the regional government's documents. (3) Generally, there are five main factors causing inconsistencies in national and regional planning in the agricultural sector that exist in Indramayu Regency and South Lampung Regency, which are policies made by regional officials, force majeure, development priorities, communication distortion, and the support of political parties. The force majeure factor (impacts of COVID-19), the communication distortion between regional and central governmental officials, and the support of political parties are all new findings of this study. These factors represent the main factors causing the inconsistency between national and regional planning in the agricultural sector. In order to improve the consistency and integration between national and regional planning on the agricultural sector, the central government (Ministry of Agriculture) and government officials in charge of the regional agricultural sectors have to collaborate by making an electronic priority scale for development planning and reinforcing regional planning capacity. This research is limited to two areas that have optimal resources, such as large areas of paddy fields and abundant air resources, which has implications for the number of programs/activities allocated to these regions by the central government. In order to be more comprehensive, the researchers suggest that further research be carried out regarding the consistency of planning at various levels in areas that have limited resources.

**Author Contributions:** W. performed data collection, data analysis, and the writing of this paper. D.S. analysed and provided manuscript correction. I.M.F. and S.S. provided the name of this manuscript. All authors have read and agreed to the published version of the manuscript.

**Funding:** We would like to thank the Ministry of Agriculture, Indramayu Regency's Department of Agriculture, and South Lampung Regency's Department of Agriculture for all the financial support to accomplish and finalize this script.

**Institutional Review Board Statement:** Not applicable.

**Informed Consent Statement:** Not applicable.

**Data Availability Statement:** All data generated or analysed during studies are currently available from the author upon reasonable request.

**Conflicts of Interest:** This study is conducted without the existence of commercial and or financial relations that can be interpreted as a potential conflict of interest.

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
