# Peer review of "Consistency of Central and Regional Planning in the Agricultural Sectors and the Factors Affecting It in Indonesia"

_sustainability, doi:10.3390/su142316297_

Round 1
Reviewer 1 Report
Very informative and applicable research.
Author Response
Response to Reviewer#1 Comments
Point 1: Very informative and applicable research
Response 1: Thank you very much. We appreciated and hopefully our manuscript can be accepted
Reviewer 2 Report
Please the Authors clarify what they mean when using the term of 'integration' in the introduction.
-
1. I think that the principal research question of the paper is regarding the affordability of the consistency analysis applied to the development of place-based programs by using qualitative methods. Please the authors specify the research question in the introduction and regarding the difference with the statistical approach (i.e: Swarat Chaudhuri, Azadeh Farzan, Zachary Kincaid, 2014, Consistency Analysis of Decision-Making Programs, POPL '14: Proceedings of the 41st ACM SIGPLAN-SIGACT Symposium on Principles of Programming Languages, 2014, Pages 555–567, https://doi.org/10.1145/2535838.2535858)
2. Please the authors specify what they consider the topic original or relevant in the field. Does it address a specific gap in the field of multi-dimensional planning in the agricultural sector?
3. Please have the authors clarify why they choose references 47 and 49 as methodologies preferred. Please have the authors indicate the possible other methodologies to complete the research (if any).
4. Please include any additional comments on the figure … that is quite different from the original reference 50 as you can see on page 3 (description of analysis) and figure 1 “Our comparative case study approach”, page. 4.
Author Response
Dear Reviewer,
We've read very carefully the suggestion and input, here we submit your feedback and please see the attachment
Response to Reviewer#2 Comments
Point 1: Please the Authors clarify what they mean when using the term of 'integration' in the introduction
Response 1: The definition of Integration in this study is a unified whole between national and regional planning (See page 1, line 41).
Point 2: I think that the principal research question of the paper is regarding the affordability of the consistency analysis applied to the development of place-based programs by using qualitative methods. Please the authors specify the research question in the introduction and regarding the difference with the statistical approach (i.e: Swarat Chaudhuri, Azadeh Farzan, Zachary Kincaid, 2014, Consistency Analysis of Decision-Making Programs, POPL '14: Proceedings of the 41st ACM SIGPLAN-SIGACT Symposium on Principles of Programming Languages, 2014, Pages 555–567, https://doi.org/10.1145/2535838.2535858)
Response 2: Thank you. The main problem in this research is the non-linearity of a program/activity carried out by the central and regional governments. Often programs instructed to the regions are not implemented, and vice versa.
We already fix the research question of this study that how consistent planning is carried out by the central and regional governments, from central documents to regional documents, and documents to implementation in the field. (please see page 2, line 90-92)
The approach taken in this study uses qualitative analysis of content and consistency, as well as a comparative case study for case studies on planning issues. Whereas in the study of Swarat, et all (2014), they assessed consistency in a computational geometrics program, this method is less relevant when applied to case study-based qualitative research on a program/activity.
Point 3: Please the authors specify what they consider the topic original or relevant in the field. Does it address a specific gap in the field of multi-dimensional planning in the agricultural sector?
Response 3: Thank you for your suggestion. The original topic in this study is use the government program to reduce the inconsistency of regional and center.
In addition, this study look at programs/activities from the government, not programs on a computing program, so that the research results of the computational program are less relevant to be used to address multi-dimensional inequalities in the agricultural sector. to strengthen it, we’ve added in the text (please see page 2-3, line 93-105).
Point 4: Please have the authors clarify why they choose references 47 and 49 as methodologies preferred. Please have the authors indicate the possible other methodologies to complete the research (if any).
Response 4: Refereces 47: RK Northon (2007) uses content analysis by measuring the number of words or phrases. Measuring the frequency and intensity of key texts for qualitative-based research, while 49: J Maxwell (2013) is an expert on qualitative research design. Maxwell emphasizes the meaning of a qualitative research by comparing or comparing existing cases according to the needs of current researchers.Regarding this matter, we’ve explained in the chapter methods of analysis line 183-191 and 211-220. Several methods have been seen by researchers, but the method with content analysis and comparative case study is the most suitable for explaining the consistency and factors that influence national and regional planning.
Point 5: Please include any additional comments on the figure … that is quite different from the original reference 50 as you can see on page 3 (description of analysis) and figure 1 “Our comparative case study approach”, page. 4
Response 5: I've completed the fix the editor wanted (please see page 3 line 125-126, page 4 line 154-155, page 6 line 247-248).

Reviewer 3 Report
I have carefully read the article and I encourage the publication of such an article.
The article has many merits but also some shortcomings that could be corrected.
The subject, of wide interest, is focused on the degree of concordance between national and local policies applied to a region in Indonesia.
Since the magazine where it was proposed for publication has a wide international circulation, the work should be anchored more strongly in the international literature. The theoretical-methodological framework should include more references to works with wide international circulation, to articles from the flow of international circulation. Also, the conclusions of the study should better highlight its importance and its international relevance. Thus, the study could be successfully used as a model for other similar researches.
The abstract reflects the content of the article.
I think the analysis methodology is ok, but especially at the end of the study there are too many citations that should be interpreted by the authors.
The structure of the article raises several problems. I think that the presentation of the study area is independent of the methodology and should be done in a separate section. Also, section 3.2.3. it cannot start with a table, especially since it is cited previously. Section 3.2.2 should be joined and 3.2.3. There cannot be a "Results and Discussions" section and another "Discussions" section. They must be brought together. From my point of view, part of the "Discussions" section would go to the conclusions.
The conclusions are too little developed in relation to the extension of the study. The importance of the study, the international relevance of the study and possible future research directions deriving from this study should complete the conclusions section.
Author Response
Response to Reviewer#3 Comments
Dear Reviewer,
We've read very carefully the suggestion and input, here we submit your feedback and please see the attachment
Kindly Regards
Co-Author and Team
Point 1: I have carefully read the article and I encourage the publication of such an article.
Response 1: We thank you for the excellent reviewer's attention, we hope to be able to the next stage
Point 2: The article has many merits but also some shortcomings that could be corrected. The subject, of wide interest, is focused on the degree of concordance between national and local policies applied to a region in Indonesia. Since the magazine where it was proposed for publication has a wide international circulation, the work should be anchored more strongly in the international literature. The theoretical-methodological framework should include more references to works with wide international circulation, to articles from the flow of international circulation. Also, the conclusions of the study should better highlight its importance and its international relevance. Thus, the study could be successfully used as a model for other similar researches
Response 2: Thank you very much for your comments. We already put the international references to sharp the study framework and methodologies with the addition 15 references (please see page 1-2 line 93-105).
We also agree with the reviewer's opinion to sharpen the conclusions by highlighting their importance and relevance so that they can be used as a model for similar research (already included in the conclusion of the last paragraph line 643-647)
Point 3: The abstract reflects the content of the article.
Response 3: Thank you very much
Point 4: I think the analysis methodology is ok, but especially at the end of the study there are too many citations that should be interpreted by the authors.
Response 4: in the concluding chapter, we do not add citations, maybe what the reviewer meant was in the discussion chapter. In the discussion section, we deliberately compare our findings with those of other researchers so as to further strengthen the discussion and novelty of the research we have obtained
Point 5: The structure of the article raises several problems. I think that the presentation of the study area is independent of the methodology and should be done in a separate section. Also, section 3.2.3. it cannot start with a table, especially since it is cited previously. Section 3.2.2 should be joined and 3.2.3. There cannot be a "Results and Discussions" section and another "Discussions" section. They must be brought together. From my point of view, part of the "Discussions" section would go to the conclusions
Response 5: We totally agree with the reviewer, section 3.2.3 is a typo, we have removed that sub-section (please see page 17 line 350-351)We also agree with the reviewer to eliminate the Results and Discussion Sections to become Results only (pelase see page 6 line 249)
Point 6: The conclusions are too little developed in relation to the extension of the study. The importance of the study, the international relevance of the study and possible future research directions deriving from this study should complete the conclusions section
Response 6: we agree with the reviewers, therefore we strengthen the conclusion by adding the importance of research, international relevance of research (15 new references) and possible future research directions derived from this research must complement, the authors have added to the conclusion of the last paragraph (please see page 25 line line 636-649 )

Reviewer 4 Report
The text, while dealing with an interesting topic, is structurally flawed.
It needs to present the research questions in a clear way.
It should more clearly refer to the planning levels that exist in the country, define concepts such as harmonisation and specialisation between spatial planning levels. Better identify which planning level has a regulatory role.
The conclusions are also insufficient. The reference only to covid and the comunication distortion between regional and central government is self-evident, in the sense that not all this analysis is needed to prove it.
Author Response
Response to Reviewer#4 Comments
Dear Reviewer,
We've read very carefully the suggestion and input, here we submit your feedback and please see the attachment
Kindly Regards
Co-Author and Team
Point 1: The text, while dealing with an interesting topic, is structurally flawed. It needs to present the research questions in a clear way.
Respose 1: We’re very thank you for your attention. The main question in this research is how consistent planning is carried out by the central and regional governments, seen from central documents to regional documents, and documents to implementation.To strengthen the presentation, we’ve followed the reviewer's suggestion by adding to line 90-92.
Point 2: It should more clearly refer to the planning levels that exist in the country, define concepts such as harmonisation and specialisation between spatial planning levels. Better identify which planning level has a regulatory role
Respose 2: The main planning issue in this country is the consistency of national planning with regional planning. Therefore, we focus more on the level of consistency and its factors. However, the reviewer's suggestion is very useful to fill the gaps in this research, we have included the reviewer's suggestion in the conclusion section on line 643-649)
Point 3: The conclusions are also insufficient. The reference only to covid and the comunication distortion between regional and central government is self-evident, in the sense that not all this analysis is needed to prove it
Response 3: Thank you for the suggestion, to make it more comprehensifve, we’ve added: 1. 15 international references (from 61-76); 2. Research power not only from the results but also the analysis used (plese see page 25 line 617-619); 3. possible future research (please see page 25 line 645-646); 4. There are actually 3 new findings, but the authors missed mentioning them, namely: The force majeure factor (impacts of Covid-19), communication distortion between regional and central governmental officials, and the support of political parties

Round 2
Reviewer 4 Report
I agree with the comments and corrections made